# Knowledge and Factors Associated with Breast Cancer Self-Screening Intention among Saudi Female College Students: Utilization of the Health Belief Model

**DOI:** 10.3390/ijerph192013615

**Published:** 2022-10-20

**Authors:** Nasser Shubayr, Rola Khmees, Ali Alyami, Naif Majrashi, Nada Alomairy, Siddig Abdelwahab

**Affiliations:** 1Department of Diagnostic Radiography Technology, College of Applied Medical Sciences, Jazan University, Jazan 45142, Saudi Arabia; 2Medical Research Centre, Jazan University, Jazan 45142, Saudi Arabia; 3College of Medicine, Jazan University, Jazan 45142, Saudi Arabia

**Keywords:** breast cancer, health belief model, breast self-examination intention, undergraduate female students, cross-sectional design

## Abstract

Breast cancer (BC) screening is vital, as it is linked to a greater likelihood of survival, more effective treatment, and better quality of life. One of the most extensively applied models for conceptualizing probable barriers and facilitators to the adoption of desired health behavior is the health belief model (HBM). This study aimed to assess the impact of health perception and knowledge on breast self-examination intention (BSE) using HBM. HBM measures specific factors (perceived susceptibility, severity, barriers, benefits, cues to action, and self-efficacy) that impact one’s intention to use BSE. Data were collected from female undergraduate students (*n* = 680) using a cross-sectional design, stratified simple random sampling, and a self-administered structured online questionnaire. We used multivariable logistic regression to evaluate our assumptions for students who intended to self-examine for BC. For demographic factors, we modified the multivariate model. Most respondents (93%) were under the age of 24 years. Female students from health colleges (48.5%), in their fourth academic year (42.6%), and from the southern region (57.6%) accounted for the majority of the sample. Respondents with a family history of BC were 9.7% of the total. Categories for age, college, region, residency, and BSE intention showed significant differences in their average knowledge scores. The survey revealed that 72.4% were aware of abnormal breast changes. Three constructs of the health belief model (perceived benefit, perceived barriers, and self-efficacy) are good predictors of BSE intention. Theory-based behavioral change interventions are urgently required for students to improve their prevention practices. Furthermore, these interventions will be effective if they are designed to remove barriers to BSE intention, improve female students’ self-efficacy, and enlighten them on the benefits of self-examination.

## 1. Introduction

Breast cancer (BC) is the most frequent type of cancer among women worldwide and has continually risen in recent years [1]. In 2018, it was projected that over 2.1 million new cases of BC would occur worldwide, accounting for nearly 1 in 4 cancer cases among women [2]. Early detection of cancer is crucial to minimize mortality [3,4]. According to the literature, cancer death rates have grown by around 14% since 2008, increasing in underdeveloped nations due to delayed diagnosis and insufficient treatment [5].

The current approaches for detecting BC are mammography, clinical breast exams, and breast self-examination (BSE) [6]. Despite global recommendations for screening, current practice is still suboptimal [7]. Although the World Health Organization (WHO) does not recommend BSE as a screening test for BC, it can raise awareness of what is normal breast and recognize changes in their breast, size, shape, skin, and nipples as soon as possible by self-examination [8], and if any abnormal changes are noticed, it alerts women and doctors to the importance of performing more advanced screening measures, particularly for women with a positive family history of BC [9]. The ethics of BC screening programs were addressed by WHO [10].

Breast screening practices are influenced by risk, benefit, and barrier perceptions via a reasoning process that incorporates personal and societal influences and attitudes [11]. Numerous methods have been developed to explain health-related behavior since the early 1950s. The health belief model (HBM) with its essential components is one of the most widely used models for conceptualizing the possible obstacles or facilitators of desirable health adoption behavior [12]. HBM’s essential elements center on individual beliefs about health conditions and are predictive of individual health-related behaviors. The model identifies the critical factors influencing health behaviors as an individual’s perceived threat of illness or disease (perceived susceptibility), belief in the severity of the consequences (perceived severity), perceived benefits of action, perceived barriers to action, exposure to elements that prompt action (cues to action), and confidence in one’s ability to succeed (self-efficacy) [13]. HBM components have been widely used to assess patterns of adopting BSE and mammography throughout the years, primarily via translations and modifications of BC’s HBM scale [14,15,16,17]. Psychometric characteristics of this scale have been examined in both Western and non-Western cultures. Numerous studies have established that HBM is a viable and reliable tool for assessing health attitudes about BC and screening strategies [15,16]. According to HBM, a woman must believe herself to be vulnerable to BC and be well-informed about the numerous other perceptions that impact attitudes and behaviors and the advantages and obstacles to engaging in BC screening [14,15,16].

Saudi Arabia is classified by the World Bank as a high-income economy and ranked 10th globally in terms of the lowest poverty rate [18]. In Saudi Arabia, BC is the most frequent type of cancer among women, with an age-standardized incidence and mortality rates of 27.3 and 7.5 per 100,000 women, respectively [19]. Furthermore, prior studies have revealed a rising tendency in the occurrence of BC among Saudi women over time [20,21,22]. Several studies addressed socio-cultural, family values, and religious beliefs as the main factors contributing to late diagnosis [23]. In addition, cultural norms (i.e., having a female doctor for clinical breast examinations and mammograms) and religious beliefs (i.e., the cause of BC is a test or punishment from God for previously committed sins) constituted barriers to early diagnosis and treatment [24,25]. Therefore, immediate interventions are required to expand awareness and encourage early detection of cancer among women.

In the present study, we examined the knowledge and factors associated with BSE intention using HBM, as a theoretical basis to explore variables affecting BSE behavior, among female college students in Saudi Arabia. 

## 2. Materials and Methods

### 2.1. Study Type and Sample

This cross-sectional study was conducted from January to March 2022. The sample size (*n* = 680) was determined by taking *p* (expected proportion) = 0.5, *z* = 1.96 (level of significance), and *d* = 0.30 (margin of error) with G*Power software, version 3.2 [26]. The stratified random sampling technique was used for this study. The population of interest for this research was female students enrolled at the Saudi Arabian universities. The university students were from five universities in various regions of the Saudi Arabia: middle, eastern, western, southern, and northern. The statistics show that Saudi women make up 51.8% of the university students in Saudi Arabia. Compared to 513,000 men, 551,000 women are pursuing bachelor’s degrees. The inclusion criteria for this study were female, at least 18 years old, presently registered as a student at a university, able to speak and write Arabic, and willing and competent to complete an online questionnaire. The data collection instrument was a web-based survey utilizing an electronic Google Form questionnaire. Student leaders from different universities were contacted to invite students from various colleges to participate in the survey. The contacts of the students’ leaders were obtained through personal contact with the academic staff from various departments of diagnostic radiography technology in the Saudi universities. The questionnaire link was distributed to students via an online social platform (WhatsApp) by the students’ leaders. All participants that met the selection criteria were recruited and all sources of statistical bias were avoided. The response rate was 98.5%. Sampling from different geographical regions was carried out to ensure external validity.

### 2.2. Ethical Approval 

The Jazan University Research Ethics Committee (Approval No. REC76/1/004) granted ethics approval prior to data collection. We followed the guidelines specified in the Declaration of Helsinki for Human Studies in this research. Participants were assured of the confidentiality of their data. The confidentiality of the participants’ personal information in this study was protected by omitting their personal information from the questionnaire. The data were only available to the research team. We obtained their informed consent to participate.

### 2.3. Data Collection Tools

Sociodemographic data, academic background, family history, BSE intention, knowledge construct [27], and the Champion’s Health Belief Model Scale [28] were used. BC knowledge was assessed with a 19-item test that measures subjects’ knowledge of BC detection and screening practices. Each correct response was scored as 1, and each false and “do not know” response was scored as 0. The total score then converted to a 0 to 10 scale, where the cut off score was assigned at 5. The Champion’s Health Belief Model Scale is well known [29], and the validity and reliability of its Arabic version have been established [30,31]. Perceived susceptibility, perceived severity, perceived benefits, perceived barriers, cues to action, and self-efficacy are the six subscales addressed using a five-point Likert-type scale. The questionnaire was then pilot tested with students from colleges outside the study’s stratum, and the phrasing of a few questions related to socio-cultural aspects was modified. The Cronbach’s alpha coefficients of the subscales were determined to be 0.82–0.93 in this study. 

### 2.4. Statistical Analysis 

The Statistical Package for Social Sciences (SPSS) version 26 software (IBM Corp., Armonk, NY, USA) was used to analyze the data. The participants’ characteristics were analyzed using descriptive statistics based on frequencies, percentages, averages, and standard deviations. The test of normality, the Kolmogorov–Smirnov test, was used. The nonparametric tests, the Mann–Whitney and Kruskal–Wallis tests, were applied to determine associations between variables due to the violation of normality. The dependent variable “BSE intention” (yes/no) was compared to a set of independent factors using logistic regression analysis. The BSE intention as the dependent variable was modeled using logistic regression analysis and the “Enter” method. Adjusted odds ratios (AORs) and their 95% confidence intervals were determined for all the independent variables. Age, academic level, region, college, knowledge score, and HBM constructs were the independent variables for the logistic regression models. Statistical significance was established as a *p*-value of 0.05.

## 3. Results

Most respondents (93%) were under the age of 24 years. Female students from health colleges (48.5%), in their fourth academic year (42.6%), and from the southern region (57.6%) accounted for the majority of the sample. Of the participants, 68.8% lived in cities, and 31.2% lived in villages and rural areas. A majority of participants (90.3%) reported no family history of breast cancer, yet 23.7% had had a family member or friend experience breast cancer. Only two women (0.3%) had ever had BC. The rate of respondents with the intention of performing BSE was 77.8% (Table 1) while 76.5% felt a greater intention when talking a great deal about BC. The overall mean knowledge on BC was 4.95 ± 2.24, with students in health-related colleges showing the highest knowledge score (5.68 ± 1.99). Categories for age, college, region, residency, and BSE intention showed significant differences in the average knowledge scores (Table 1).

The knowledge of BC screening is shown in Table 2. Around half of the participants (55.7%) knew that the vast majority of breast lumps tend to first be detected by the women themselves while 24.3% knew that postmenopausal women should perform a monthly BSE, and 72.4% were aware of abnormal breast changes. 

The mean scores obtained from the participants for the HBM constructs are given in Table 3. The means for perceived benefits and perceived susceptibility were the highest and lowest, respectively. There was a statistically significant (*p* < 0.01) association between BSE intention and perceived benefits, barriers, and self-efficacy. 

Because the scores for the majority of constructs did not match the normalcy assumption, we used Spearman’s correlation to determine the relationship between the HBM constructs (Table 4). Self-efficacy was positively and significantly linked with BC knowledge (*r* = 0.176, *p* < 0.01). The results indicated that perceived barriers (*r* = −0.141, *p* = 0.01) and perceived benefits (*r* = −0.112, *p* < 0.01) were negatively and significantly linked with BC knowledge, whereas perceived susceptibility, perceived severity, and cues to action were not.

The logistic regression findings presented in Table 5 reveal no significant age differences. Academic level and region did not show any significant associations with BSE intention. Three constructs of the health belief model (perceived benefits, perceived barriers, and self-efficacy) were good predictors of BSE intention. The construct “self-efficacy” significantly predicted the acceptance of BSE intention, with an AOR = 1.83 (95% CI: 1.29–2.58, *p* < 0.01). Perceived susceptibility, perceived severity, and action cues did not significantly predict BSE intention. 

## 4. Discussion

This study was designed to investigate the knowledge and factors associated with BSE intention among Saudi female college students using HBM. HBM is a theoretical framework that can guide programs aimed at promoting health and preventing diseases. It is used to describe and forecast changes in individual health behaviors. It is a widely used model for analyzing health behaviors in the health sector and in psychological research to explain and predict human behavior. Two previous studies in Saudi Arabia used HBM to predict current behavior among study participants but not the behavior the participants intend to take [30,31]. However, in this study, BSE intention was designed as a dependent variable to be explained by demographic factors and health belief models, in line with a study conducted in Indonesia in which intention to perform BSE was used as a dependent variable [32]. The designation of intention as a dependent variable has also been used in other studies, such as the intention to take the COVID-19 vaccine [33]. In addition, the region in which this current study was conducted was descriptive [34] of quasi-experimental BC [35] studies.

The percentage of respondents who showed BSE intention in this study was 77.8%. It has been reported in the literature that the percentages of women performing BSE in developing countries range from 17% to 50% [36,37,38,39]. In studies carried out at a local level in Saudi Arabia, the percentages of women performing BSE ranged from 20% to 61% [40,41,42,43,44]. This aligns with reports in other nations, such as Nigeria, Austria, Sweden, and Egypt [45,46,47,48].

The overall mean of knowledge of BC was 4.95 (± 2.24). Students in health-related colleges scored the highest mean knowledge, which is in line with a previous study from India [49]. Categories for age, college, region, residency, and BSE intention showed a significant difference in their average knowledge scores (Table 1). This finding is consistent with the findings of two previous Saudi Arabian studies. The first study, conducted in Jeddah, concluded that the majority of the participants (57.5%) were aware of a family history of BC and having a close relative with the disease as an established risk factor for the disease. Additionally, 41.0% and 35.5% of the participants were aware of alcohol consumption and hormone replacement therapy (HRT), respectively, as additional risk factors for BC [43]. The second study, conducted in Riyadh, identified heredity and HRT as common BC risk factors, as perceived by Riyadh women [44]. The older age groups reported the highest knowledge scores. This finding is consistent with the published literature in which older age groups were found to be more educated about BC [50]. 

Prevention requires an understanding of risk factors. Our study’s findings indicate that women have a low-to-moderate level of knowledge about BC, regardless of their academic degree, academic year, or family history of BC. The same pattern was observed in terms of mammography knowledge and practice. These women may derive their knowledge from a variety of sources.

Despite the demonstrated benefits of screening for early detection of BC, collective harms also were documented, where differences in balancing the benefits and harms have led to differences among major guidelines. In addition, screening recommendations based on evidence depend on factors such as the risk (average risk vs. high risk) and screening methods [51,52]. However, according to Houssami, “the trade-off between the benefit and the collective harms of BC screening, including false-positives and overdiagnosis, is more finely balanced than initially recognized, however the snapshot of evidence presented on overdiagnosis does not mean that breast screening is worthless” [53]. Therefore, it is important that young women, since puberty, are aware of the possibility of changes in breast tissue and should be trained by experts in the examination technique. Given the frequency of breast changes, most of them are not cancer. If the woman finds a change in the breast, she must consult a physician. The physician could initially recommend a diagnostic mammography [8].

The means for perceived benefits and perceived susceptibility were the highest and lowest, respectively. There was a statistically significant (*p* < 0.01) association between BSE intention and perceived benefits, perceived barriers, and self-efficacy. Understanding and improving the factors that influence women’s behavior regarding BC screening will aid in BC prevention. Identifying these factors also aids researchers in developing and implementing effective behavioral change interventions [54]. Additionally, behavioral change interventions that focus on theory-centered determinants are likely to be more effective [55,56]. For example, the findings from an Iranian study indicated that women who engaged in BSEs had significantly higher perceived benefits and self-efficacy than those who did not engage in BSEs. Additionally, those who had performed BSEs reported fewer perceived barriers [55]. In South Korea, they discovered that perceived benefits, perceived susceptibility, and perceived barriers were the primary predictors of mammography screening using two behavior change models: the Health Belief Model and the Transtheoretical Model [57]. Self-efficacy refers to a person’s belief in their capacity to take action or accomplish a specific task. People seldom attempt to acquire new behaviors unless they are confident of their ability to do so. A person who believes that changing their behavior is beneficial (perceived benefit) but is skeptical of their capacity to do so is unlikely to try to change their lifestyle. In other words, even if a person feels that adopting healthier behaviors would have considerable advantages, they are unlikely to alter their existing habits if they have doubts about their ability to overcome the hurdles to change. Encouragement, training, and other forms of support can help boost self-efficacy [58].

After adjusting for age, academic level, region, educational background, and knowledge score, multivariate logistic regression was used to understand the relationship between BSE intention and HBM constructs and estimate AOR. The logistic regression findings revealed no significant association between BSE intention and age, academic level, region, and educational background. The knowledge score had a positive impact on BSE intention [AOR = 1.30 (95% CI: 1.18–1.43, *p* < 0.01)]. Similar results were observed in Ethiopia [59,60], Turkey [61], Palestine [62], Ghana [63], Vietnam [64], and other studies in Saudi Arabia [65]. The findings of the present study showed that only three constructs of the health belief model (perceived benefit, perceived barriers, and self-efficacy) are good predictors of BSE intention. A study conducted on Grenadian [66] and Turkish [67,68] women proved that perceived benefit significantly predicts BSE. BSE among nonmedical female students in Ghana was determined by self-efficacy and perceived barriers [63].

### Limitations of this Study

Our study is not without limitations. First, the study relied on cross-sectional data, which may have restricted the interpretation of any causative relationship. Second, recollection bias may have altered the respondents’ responses. When research participants are more or less likely to remember information on exposure and link it to their result status, or when they are more or less likely to recall information on their outcome based on their exposure, this is known as memory bias. Finally, this study’s sample methodology and the differential response rates among chosen universities may restrict this study’s generalizability. Despite these constraints, our research demonstrates the intention among university students to perform BSE.

## 5. Conclusions

The rate of respondents who showed BSE intention in this study was 77.8%. Given that participants in the present study group were well educated, these rates may be considered moderately low. The previous literature has found that the most important barrier to developing a behavior is the perception of the barrier and that this perception can be changed by education, counseling, and approaches aimed at increasing access to health services; thus, as the perception of a barrier decreases, the perception of a benefit increases [30]. In the present study, the perception of a barrier was related to BSE intention. Self-efficacy and perceived benefits also affected BSE intention. Encouragement, training, and other forms of support can help boost self-efficacy. 

Health professionals working in primary healthcare facilities should strive to raise women’s knowledge and awareness of BC and screening procedures, and to educate them about the national BC screening program and urge them to participate. Women should be encouraged to participate in and enhance their BSE abilities. Future research should conduct focus groups or in-depth interviews to ascertain the factors that contribute to the perception of hurdles to performing BSE and having routine mammography. Relevant actions should be devised to address the factors discovered. Primary healthcare facilities should arrange education and consultation programs aimed at raising motivation by addressing the knowledge of the susceptibility, severity, and benefits of screening behaviors while removing barriers. Community-based research should be conducted to measure BC screening practices and the link between BC screening behaviors and health attitudes in women. Future research should look at not just health attitudes but also psychological and health care use characteristics that may be connected with screening. We recommend that future research on BC screening practices use health institution records rather than self-reporting.

## Figures and Tables

**Table 1 ijerph-19-13615-t001:** Knowledge scores according to the demographic characteristics.

Variables	N	%	Knowledge Score		*p*-Value
			Mean	SD	
Age (Years)					
18–19	184	27.1	4.46	2.13	0.002 *
20–24	448	65.9	5.08	2.27	
More than 25	48	7.0	5.71	2.19	
Academic year					
First Year	161	23.7	4.26	2.26	0.363 *
Second Year	122	17.9	4.53	2.14	
Third Year	107	15.7	4.96	1.99	
Fourth Year and Above	290	42.6	5.50	2.22	
College					
College of Science	56	8.2	4.68	2.17	0.00 *
Health related colleges	330	48.5	5.68	1.99	
College of Arts and Human Sciences	43	6.3	3.67	2.04	
College of Business	53	7.8	4.00	2.27	
College of Education	24	3.5	4.38	2.39	
Engineering and Technology	46	6.8	4.11	2.27	
Computer and IT	54	7.9	4.44	2.27	
Law and Islamic regulations	21	3.1	4.38	1.60	
Preliminary Year	53	7.8	4.42	2.53	
Region					
Middle Region	90	13.2	5.10	2.31	0.001 *
Eastern Region	33	4.9	4.97	1.86	
Western Region	134	19.7	4.59	2.32	
Southern Region	392	57.6	5.00	2.22	
Northern Region	31	4.6	5.39	2.23	
Residency					
Urban	468	68.8	4.83	2.25	0.044 **
Rural	212	31.2	5.21	2.18	
Family history					
Yes	66	9.7	5.59	2.08	0.162 **
No	614	90.3	4.88	2.25	
Family member or friend experienced BC					
Yes	161	23.7	5.29	2.13	0.068 **
No	519	76.3	4.85	2.26	
Previously diagnosed with BC					
Yes	2	0.3	4.50	0.71	NA
No	678	99.7	4.95	2.24	
BSE intention					
No	151	22.2	3.68	2.36	0.000 **
Yes	529	77.8	5.31	2.07	
Difference when talking about BC					
A great deal	520	76.5	5.24	2.07	0.00 *
Some difference	62	9.1	4.79	2.27	
Little or no difference at all	14	2.1	5.00	2.54	
Don’t know	84	12.4	3.26	2.44	
Total	680	100	4.95	2.24	

* *p*-value based on the Kruskal–Wallis test. ** *p*-value based on the Mann–Whitney U test.

**Table 2 ijerph-19-13615-t002:** Breast self-examination items of the participants.

Variables	N	%
Most breast lumps are found by		
Women themselves (Correct)	379	55.7
Physician	41	6.0
Mammography	151	22.2
I don’t know	109	16.0
If you are postmenopausal, how often should you do breast self-examination?		
Once every one month (Correct)	165	24.3
Once every three months	262	38.5
I don’t know	253	37.2
When feeling (palpating) the breast, you should:		
Use the pads of your fingers (Correct)	155	22.8
Use the tips of your fingers	364	53.5
Don’t know	161	23.7
Abnormal breast change		
Discharge	4	0.6
Lump, hard knot, or thickening	84	12.4
Dimpling of skin	8	1.2
All of the above (Correct)	492	72.4
None of the above	5	0.7
Don’t know	87	12.8
At what age should a woman begin breast self-examination?		
20 (Correct)	333	49.0
30	105	15.4
35	115	16.9
Don’t know	127	18.7

**Table 3 ijerph-19-13615-t003:** Mean scores on the Health Belief Model subscale.

HBM Constructs	No of Items	Min	Max	Mean (SD)	BSE Intention Mean (SD)		*p*-Value *
					Yes	No	
Perceived Susceptibility	6	1	5	2.22 (0.73)	2.23 (0.70)	2.17 (0.83)	0.234
Perceived Severity	12	1	5	2.23 (0.73)	2.22 (0.71)	2.29 (0.81)	0.299
Perceived Benefits	11	1	5	3.19 (0.90)	3.25 (0.87)	3.01 (0.98)	**0.003**
Perceived Barriers	17	1	5	2.36 (0.67)	2.32 (0.66)	2.50 (0.71)	**0.002**
Cues to Action	8	1	5	2.51 (0.70)	2.52 (0.69)	2.48 (0.79)	0.525
Self-Efficacy	12	1	5	2.29 (0.74)	2.36 (0.74)	2.05 (0.69)	**0.000**

* *p*-value based on the Mann–Whitney U test. Bold font indicates statistical significance.

**Table 4 ijerph-19-13615-t004:** Spearman correlation among the HBM constructs and knowledge score.

	Knowledge Score	Perc Susceptibility	Perc Severity	Perc Benefits	Perc Barriers	Cues to Actio	Self-Efficacy
Knowledge score	1.000	−0.054	−0.038	−0.112 **	−0.141 **	−0.002	0.176 **
Perc susceptibility		1.000	0.507 **	0.067	0.320 **	0.124 **	0.238 **
Perc severity			1.000	0.121 **	0.509 **	0.185 **	0.251 **
Perc benefits				1.000	0.181 **	0.226 **	0.160 **
Perc barriers					1.000	0.198 **	0.163 **
Cues to action						1.000	0.426 **
Self-efficacy							1.000

** Spearman’s rho correlation is significant at the 0.01 level (2-tailed). Perc: perceived.

**Table 5 ijerph-19-13615-t005:** Multivariate logistic regression analysis of independent predictors of the BSE intention.

Variables	*p*-Value	Adjusted OR	95% C.I. for OR	
			Lower	Upper
Age (Years)				
18–19 (ref)				
20–24	0.13	0.62	0.34	1.14
More than 25	0.55	0.74	0.28	1.97
Academic Level				
First Year (ref)				
Second Year	0.36	1.36	0.71	2.61
Third Year	0.36	1.43	0.67	3.04
Fourth Year and Above	0.21	1.57	0.78	3.15
Region				
Middle Region (ref)				
Eastern Region	0.44	0.68	0.26	1.80
Western Region	0.46	1.31	0.64	2.65
Southern Region	0.74	1.11	0.61	2.03
Northern Region	0.58	1.40	0.43	4.52
College				
Non-medical colleges (ref)				
Medical colleges	0.01	0.56	0.36	0.88
Knowledge Score	0.00	1.30	1.18	1.43
HMB covariates				
Perc Susceptibility	0.09	1.34	0.96	1.87
Perc Severity	0.27	0.82	0.57	1.17
Perc Benefit	0.01	1.42	1.09	1.85
Perc Barriers	0.01	0.60	0.42	0.87
Cues to Action	0.14	0.78	0.56	1.08
Self-Efficacy	0.00	1.83	1.29	2.58

## Data Availability

The data is available upon request.

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
