# Peer review of "Knowledge and Factors Associated with Breast Cancer Self-Screening Intention among Saudi Female College Students: Utilization of the Health Belief Model"

_ijerph, 2022, doi:10.3390/ijerph192013615_

Round 1

Reviewer 1 Report

Please, see the attached document.

Author Response

Thank you for your valuable comments that improved the paper significantly. We have addressed all your comments and highlighted changes in the revised draft. Please see the attached. 

Reviewer 2 Report

The authors assessed the knowledge and factors associated with breast cancer screening intention and utilization of the health belief model of Saudi female college students.

The Saudi female college students are a unique population in terms of both social and religious specialities. While, the authors do not address such specialities appropriately in both the Introduction or the Discussion. 

Author Response

We thank you for your valuable comments that improved our paper significantly. We addressed the social and cultural aspects in the introduction and highlighted the text in the revised draft. 

Round 2

Reviewer 1 Report

The manuscript was significantly improved. Nevertheless, there are details that should be clarified. Please, see the attached document with my observations.

Author Response

We thank the reviewer for his valuable comments that improved our paper significantly. Point to point response is attached. 

Reviewer 2 Report

The revisions are accepted.

Author Response

We thank the reviewer for accepting our paper.